materials science/nanotechnology

$Co_xSn_{(1-x)}O_2$, $IrO_2$, SPE water electrolyzer, oxygen evolution reaction

**Authors for correspondence:**
Gang Chen
e-mail: chengang@hnu.edu.cn
Hong Lv
e-mail: lvhong@tongji.edu.cn

This article has been edited by the Royal Society of Chemistry, including the commissioning, peer review process and editorial aspects up to the point of acceptance.

# Mesoporous $Co_xSn_{(1-x)}O_2$ as an efficient oxygen evolution catalyst support for SPE water electrolyzer

## Gang Chen[1], Jiakun Li[1,3], Hong Lv[2,3], Sen Wang[2,3], Jian Zuo[2,3] and Lihua Zhu[1]

[1]College of Materials and Engineering, Hunan University, Changsha, Hunan 410082, People's Republic of China
[2]School of Automotive Studies, and [3]Clean Energy Automotive Engineering Center, Tongji University, Shanghai 201804, People's Republic of China

GC, 0000-0002-0186-9766

SPE water electrolysis is a promising method of hydrogen production owing to its multiple strengths, including its high efficiency, high product purity and excellent adaptability. However, the overpotential of the oxygen evolution reaction process and consumption of Ir during charging in SPE water electrolysis will inevitably result in large energy loss and then high cost. Under these circumstances, we propose a novel $40IrO_2/Co_xSn_{(1-x)}O_2$ ($x = 0.1, 0.2, 0.3$) anode catalyst, where the $Co_xSn_{(1-x)}O_2$ support is synthesized by a hydrothermal method and $IrO_2$ is synthesized by a modified Adams fusion method. After modifying the component of $Co_xSn_{(1-x)}O_2$, the $40IrO_2/Co_xSn_{(1-x)}O_2$ exhibits an increased specific surface area, electrical conductivity and surface active sites. Moreover, a single cell is fabricated by Pt/C as cathode catalyst, $40IrO_2/Co_xSn_{(1-x)}O_2$ as anode catalyst and Nafion 117 membrane as electrolyte. The $40IrO_2/Co_{0.2}Sn_{0.8}O_2$ exhibits the lowest overpotential (1.748 V at 1000 mA cm$^{-2}$), and only 0.18 mV h$^{-1}$ of voltage increased for 100 h durability test at 1000 mA cm$^{-2}$. Consequently, $Co_xSn_{(1-x)}O_2$ is a promising anode electrocatalyst support for an SPE water electrolyzer.

# 1. Introduction

Hydrogen is regarded as one of the promising solutions for developing clean energy and solving the thorny environmental problems present on the Earth [1]. Water splitting for hydrogen generation is a major component of modern clean energy technologies [2], such as water-alkali electrolyzers [3], solid polymer electrolyte (SPE) water electrolyzers [4] and

photocatalytic/photo-electrochemical water splitting [5–8]. SPE water electrolyzers offer us an effective and simple method to produce hydrogen through reusing the surplus electric power generated by renewable energy (such as wind and photovoltaic power) [9]. As a result, SPE water electrolyzers have gained a lot of research attention [10–12]. Nevertheless, the conspicuous weakness of SPE water electrolysis should not be ignored, including the majority increases of the activation overpotential in the oxygen evolution reaction (OER) process [13]. Although some highly active and stable noble metal-based catalysts have been developed, such as $RuO_2$ and $IrO_2$ for OER, these materials are still far from large-scale application because of their high cost and scarcity. Therefore, multiple studies have been devoted to develop novel, highly efficient and low-cost catalysts (e.g. $La_2NiMnO_6$ [14], Ni-Fe-layered double hydroxide [15] and Ternary Ni−Co−Fe blue analogue [16]). However, these new catalysts too have drawbacks, such as a complicated preparation process and poor durability. Additionally, many research efforts have been made to reduce the amount of these noble metals, such as designing their bulk structural parameters (i.e. grain size, morphology, and dimensions) [17–19], tailoring composition (such as introducing foreign elements or oxides into the structure of noble metal catalysts) [20], and adopting supports (i.e. carbon-based [21], $SnO_2$ [22], $TiO_2$ [10], TiC [23]).

It is well known that an appropriate support will favour a noble metal-based material to achieve a better dispersion and greater surface area, which not only reduces the usage of noble metal but also maintains high activity for the catalysts [23]. As supports, carbon-based materials have recently received a lot of attention due to their high specific surface area and excellent electric conductivity. Unfortunately, carbon-based materials are easily electrochemically oxidized at potential above 0.206 V versus SHE [24], leading ultimately to the unsustainability of carbon-based supports in SPE water electrolyzer [25]. Therefore, the development of a highly stable and active support for the OER is still a research focus. Tin oxide ($SnO_2$), as a corrosion-resistant support, has been reported to promote the dispersion of noble metal-based materials and provide more surface active sites of catalysts [26]. Wang *et al.* reported that $SnO_2$ in the $IrO_2/SnO_2$ catalyst could act as a Brønsted base and accept protons from the $IrO_2$ sites, which is favourable to the enhancement of catalytic performance of $IrO_2/SnO_2$ catalyst [22]. Unfortunately, the current reported OER activity of $SnO_2$-based supports is still not significant because of the poor electronic conductivity of $SnO_2$. Heteroatom doping is a method of enhancing the catalytic activities of catalysts due to the charge delocalization mechanism [27]. Co is a transition element with incomplete electron shell and possesses interesting catalytic properties [28–30]. As far as we know, the effect on the electrocatalytic activity of $SnO_2$ with varying Co content doped as support for $IrO_2$ has been rarely studied.

In this study, $IrO_2$ supported by mesoporous $Co_xSn_{(1-x)}O_2$ ($x = 0.1, 0.2, 0.3$) as an anode catalyst for SPE water electrolysis has been exploited. $Co_xSn_{(1-x)}O_2$ ($x = 0.1, 0.2, 0.3$) is successfully synthesized by a hydrothermal method and $IrO_2$ is synthesized by using a modified Adams fusion method [4,31]. The structure, morphology, specific surface area, electrical conductivity and surface active sites of the prepared samples have been thoroughly determined by various characterization techniques. The electrocatalytic activity of the prepared samples as anode catalysts in single cells is also tested. The prepared samples with Co doping show low overpotential (1.748 V at 1000 mA cm$^{-2}$) and excellent stability.

# 2. Experimental section

## 2.1. Preparation of Co-doped SnO$_2$ support

The supports of $Co_xSn_{(1-x)}O_2$ ($x = 0, 0.1, 0.2, 0.3$) were synthesized by the hydrothermal method and then processed by heat treatment. One gram of hexadecyl trimethylammonium bromide (Sinopharm, Shanghai, China) was dissolved in a mixture of 20 ml of ethanol (Sinopharm) and 20 ml distilled water. Then, significant amounts of tin chloride (Sinopharm) and cobalt acetate (Sinopharm) (molar ratio ($Sn^{2+} + Co^{2+}$): CTAB = 1 : 1.05) were added to the above solution, and continuously stirred to get a homogeneous solution. A certain amount of ammonia water (Sinopharm) was added dropwise to the solution under vigorous stirring at room temperature. The Ph value of the solution was maintained at 9 and under stirring for 2 h. After it was stirred, the mixture was transferred to a stainless Teflon-lined 100 ml autoclave and kept at 180°C for 24 h in an oven. The resulting yellow precipitate was collected by centrifugation, washed with distilled water and ethanol several times to remove the impurities, and then dried in a vacuum oven at 60°C. The dried samples were processed under 350°C for 5 h in a muffle furnace. After cooling to room temperature, $Co_xSn_{(1-x)}O_2$ nanoparticles were finally obtained.

## 2.2. Preparation of supported catalysts

In this work, the $IrO_2$ loading was applied as 40 wt %, 0.196 g chloroiridic acid (Hesen, Shanghai, China), 5 g of ultrafine sodium nitrate (Sinopharm) and 0.12 g as-prepared support powder were added into 10 ml of isopropyl alcohol (Sinopharm), which was stirred to obtain uniform suspension. Then the suspension was ground for 6 h by planetary ball milling. The obtained slurry was dried in a vacuum oven at 60°C, and treated at 400°C for 1 h in a muffle furnace with a heating rate of 5°C $min^{-1}$. After heat treatment, the powder was washed in turn with 0.1 M HCl, distilled water, and ethanol to eliminate residual impurities, and dried in a vacuum oven at 70°C overnight. The resulting material was denoted as $40IrO_2/Co_xSn_{1-x}O_2$ ($x = 0.1, 0.2, 0.3$). The preparatory method of unsupported $IrO_2$ was similar to that of the $40IrO_2/Co_xSn_{1-x}O_2$ samples.

## 2.3. Physical characterization

X-ray diffraction (XRD) was performed to obtain the crystal structure and phase purity of all the prepared samples using a Bruker D8 Advance X-ray diffractometer (Bruker, Karlsruhe, Germany) with a Cu-Ka radiation source ($\lambda = 0.154056$ nm). Transmission electron microscopy (TEM) and high-resolution TEM (HR-TEM) images were carried on a JEOL 2010F microscope (JEOL, Tokyo, Japan). The specific surface area and pore size distribution of the as-prepared samples were recorded with the measurement of nitrogen adsorption isotherm at 77 K using a Micromeritics ASAP 2020 analyzer (Micromeritics, Norcross, Georgia, USA). Electrical conductivity measurements were carried out on cylindrical pellets compressed from the powder samples at 30 MPa between two copper electrodes. The substrate area was restricted to 1 $cm^2$ while the thickness of the pellet was measured by a Vernier caliper. The value of resistivity was immediately measured by a JG-ST2258A resistivity tester (Jingge Electronic, Suzhou, Jiangsu, China) by inputting the thickness-area ratio as a parameter, followed by conversion to conductivity.

## 2.4. Electrochemical characterization

The half-cell electrochemical evaluation of different samples was investigated by a three-electrode measurement in the $N_2$-saturated 0.5 M $H_2SO_4$ electrolyte. A reversible hydrogen electrode (RHE) and a platinum wire acted as the reference and counter electrode, respectively. The working electrode was prepared by a catalyst layer coating on the glassy carbon disc (GCE, 5.6 mm in diameter). Briefly, the catalyst layer was fabricated as follows: 12.46 mg of $40IrO_2/Co_xSn_{1-x}O_2$ powder was dispersed in 2 ml methanol/Nafion (50 : 1, wt.%) mixed solution and uniform ultrasound solution was obtained by ultrasound. The loadings of all samples on glassy carbon were controlled at 0.1 mg $cm^{-2}$. Cyclic voltammetric (CV) measurements were performed on a CHI 760E Electrochemical Workstation at a scanning rate of 50 mV $s^{-1}$ between 0.05 and 1.35 $V_{RHE}$. The potential range of the linear sweeping voltammetry (LSV) curve was from 1.4 to 1.65 V versus RHE at a scan rate of 50 mV $s^{-1}$, and the rotation rate of the working electrode was 1600 rpm.

The membrane electrode assemblies (MEAs) were prepared by the spray method and assembled using a Nafion117 membrane (DuPont, Wilmington, Delaware, USA) adopted as an SPE, the prepared samples were used as anodic electrocatalysts and a commercial 60 wt.% Pt/C (Johnson Matthey, London, UK) catalyst acted as cathodic electrocatalyst. Prior to the assembly, the Nafion117 membrane was cleaned by $H_2O_2$ solution, distilled water, and $H_2SO_4$ solution at 80°C for 1 h for each step. The sprayed catalyst inks were fabricated by the mixture of the obtained samples, Nafion solution, isopropyl alcohol and deionized water, and sonicated for 2 h to get a homogeneous suspension. The Nafion loading on each side of the membrane was maintained at 25 wt.%. The noble metal Ir and Pt loading on the membrane was 2.5 mg $cm^{-2}$ for the anode and 0.5 mg $cm^{-2}$ for the cathode, respectively. Finally, the MEAs (with an effective area of 3.645 $cm^2$) were assembled into a home-made single cell water electrolyzer, as shown in the schematic diagram in scheme 1. Ti mesh and plates were made as current collector and bipolar plate for the anode side, respectively. Carbon paper and Ti plates were used as current collector and bipolar plate for the cathode, respectively. Deionized water with a preheated 80°C temperature was pumped by a peristaltic pump to the anode compartment. At atmospheric pressure, the polarization curves of the single cells were measured by a Motech LPS305 programmatic DC power supply. The electrochemical impedance spectrums (EIS) for the single cells were conducted at 0.3 A $cm^2$ in the frequency range of 0.1 Hz to 10 kHz

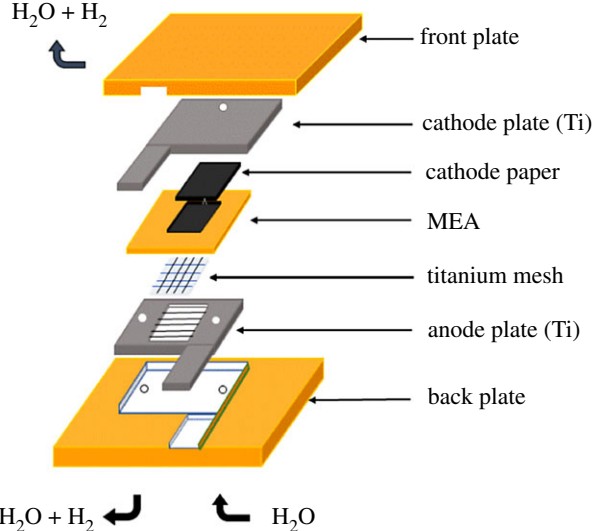

**Scheme 1.** Schematic of the SPE water electrolyzer structure.

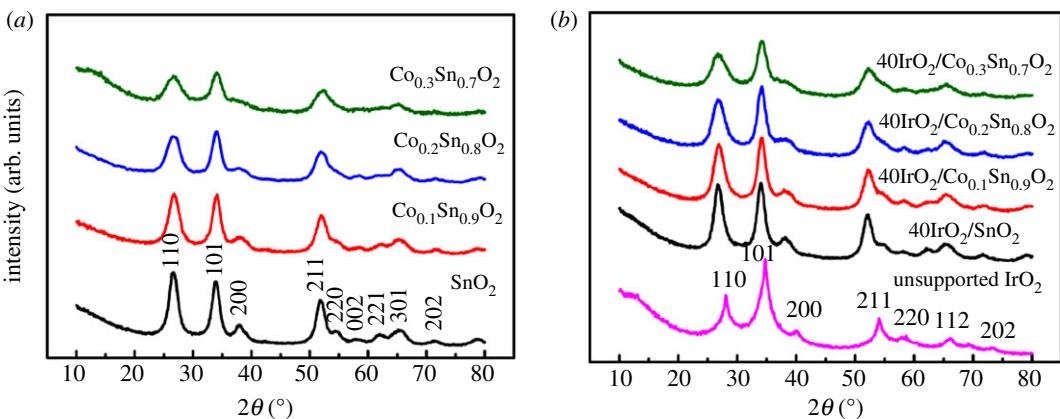

**Figure 1.** XRD patterns of the (*a*) $Co_xSn_{1-x}O_2$; (*b*) $40IrO_2/Co_xSn_{1-x}O_2$ ($x = 0$, 0.1, 0.2, 0.3) and unsupported $IrO_2$.

(amplitude $= 80$ mV) and recorded with a Solartron Analytical 1260 impedance analyzer coupled to a Solartron Analytical 1287 potentiostat.

# 3. Results and discussion

## 3.1. Physical characterization

The XRD patterns of the prepared samples are shown in figure 1. Figure 1*a* shows that all the diffraction peaks of $SnO_2$ are matched well with the characteristic peaks of tetragonal rutile structure (JCPDS 41−1445) [32]. The diffraction peaks located at approximately 26.6°, 33.9°, 37.9°, 51.8°, 54.7°, 57.8°, 62.6°, 65.9°, 71.3° and 78.7° represent the (110), (101), (200), (211), (220), (002), (221), (301), (202) and (321) planes, respectively. It is noted that the diffraction peaks of $SnO_2$ doped with varying Co content show similar shapes to the pure $SnO_2$, and no second phase about Co is detected, which confirms that Co ions were successfully doped into $SnO_2$ [33]. Figure 1*b* displays the XRD patterns of the unsupported $IrO_2$ and $40IrO_2/Co_xSn_{1-x}O_2$ ($x = 0$, 0.1, 0.2, 0.3). The typical peaks of the unsupported $IrO_2$ could be matched well with the tetragonal rutile structure [31]. After $IrO_2$ supported on $Co_xSn_{1-x}O_2$, the $40IrO_2/Co_xSn_{1-x}O_2$ exhibit similar shapes with $Co_xSn_{1-x}O_2$, implying that the loaded $IrO_2$ will not affect the crystal structure of $Co_xSn_{1-x}O_2$ by our presented modified Adams fusion treatment. The lattice parameters (a = b, c) of $SnO_2$ and $Co_xSn_{1-x}O_2$ (as listed in table 1) decrease with the increasing Co-doping concentration. This is because the radius of $Co^{2+}$ (0.072 nm) is smaller than that of $Sn^{4+}$ (0.083 nm) at a coordination number of 6 [34]. Furthermore, the

**Table 1.** The lattice constant, grain sizes, BET surface area and BJH adsorption average pore diameter results of $Co_xSn_{1-x}O_2$ ($x = 0$, 0.1, 0.2, 0.3).

| samples | lattice constant | | grain size (D) nm | BET surface area ($m^2 g^{-1}$) | BJH adsorption average pore diameter (nm) |
| | a = b (Å) | c (Å) | | | |
| --- | --- | --- | --- | --- | --- |
| $SnO_2$ | 4.7367 | 3.1908 | 8.28 | 72.19 | 7.88 |
| $Co_{0.1}Sn_{0.9}O_2$ | 4.7194 | 3.1849 | 7.08 | 104.01 | 8.48 |
| $Co_{0.2}Sn_{0.8}O_2$ | 4.7084 | 3.14430 | 6.54 | 131.92 | 10.59 |
| $Co_{0.3}Sn_{0.7}O_2$ | 4.6918 | 3.1273 | 5.52 | 132.22 | 7.47 |

grain sizes of all the obtained samples were calculated using the by Debye−Scherrer equation, $L = K\lambda/(\beta \cos\theta)$, where K is the constant (0.89), $\lambda$ is the wavelength of the X-ray radiation (Cu $K\alpha$ = 0.15406 nm), $\beta$ is the line width at half maximum height and $\theta$ is the diffracting angle. The calculated grain sizes of all the prepared samples are listed in table 1. The grain sizes of $Co_xSn_{1-x}O_2$ ($x = 0$, 0.1, 0.2, 0.3) gradually decrease from 8.28 nm to 5.52 nm with the Co content increasing, which may be attributed to the fact that the highly Co doped induced a segregation at the grain boundaries, which leads to a decrease in the grain size [35].

The morphologies and particle sizes of the $40IrO_2/Co_xSn_{1-x}O_2$ ($x = 0$, 0.1, 0.2, 0.3) and unsupported $IrO_2$ were characterized by the TEM images. As shown in figure 2a, the prepared $SnO_2$ is in an irregular shape and the average particle size is approximately 10.2 nm. Figure 2b shows that the unsupported $IrO_2$ exhibits a quasi-spherical shape and serious aggregation of nanoparticles with a broad particle size distribution, which would result in a poor catalytic activity. In contrast to the unsupported $IrO_2$, the $IrO_2$ nanoparticles supported on $Co_xSn_{1-x}O_2$ present a quasi-spherical shape with sub 3 nm sizes as the darker dots in figure 2c−f. Meanwhile, $IrO_2$ nanoparticles are observed to be well-dispersed on the $Co_xSn_{1-x}O_2$ supports. It is noted that the particle sizes of $Co_xSn_{1-x}O_2$ supports decrease gradually with the increase of the Co-doped content, which is in agreement with the XRD results and previous report [34]. The reduced $Co_xSn_{1-x}O_2$ particle sizes would expose more surface area for the dispersion of $IrO_2$ and avoid the serious aggregation of $IrO_2$ nanoparticles. Therefore, the catalytic activities of prepared samples could be anticipated to enhance with the Co-doped content. Figure 3 exhibits the HR-TEM images of $SnO_2$ and $40IrO_2/Co_{0.2}Sn_{0.8}O_2$. The HR-TEM image in (figure 3a) confirms the $SnO_2$ particles present an irregular shape with a mean particle size of about 10.2 nm. The lattice fringe is about 0.335 nm, corresponding to the (110) planes of $SnO_2$. The HR-TEM image of figure 3b reveals the $IrO_2$ in darker dots with the mean particle size of about 2.25 nm, and it disperses evenly on the surface of $Co_{0.2}Sn_{0.8}O_2$. The lattice fringes are about 0.258 and 0.335 nm corresponding to the (101) plane of $IrO_2$ and (110) plane of $Co_{0.2}Sn_{0.8}O_2$, respectively. Additionally, the presence, relative amount and homogeneous distribution of the Ir and Co elements in these samples were verified by EDX mapping (electronic supplementary material, figure S1−S5). According to the obtained results, the atomic ratio of Co/Sn is around 8.1 for $Co_{0.1}Sn_{0.9}O_2$, 4.0 for $Co_{0.2}Sn_{0.8}O_2$ and 2.7 for $Co_{0.3}Sn_{0.7}O_2$, which is consistent with the designed cobalt content.

It is well known that the specific surface areas play an important role in the catalytic activity of a catalyst. Thus, to identify the impact of Co doping and $IrO_2$ loading on the specific surface areas and pore size of $SnO_2$, nitrogen adsorption−desorption measurements were carried out. Figure 4 and electronic supplementary material, figure S7 represent the $N_2$ adsorption−desorption isotherms of the $Co_xSn_{1-x}O_2$ and $40IrO_2/Co_xSn_{1-x}O_2$ ($x = 0$, 0.1, 0.2, 0.3), the inset are the corresponding Barrett−Joyner−Halenda (BJH) pore size distribution curves. Typical Langmuir type IV with an inherent hysteresis loop at relative high pressure is detected for all samples, which suggests that the pores between the nanoparticles are mainly constructed by mesoporous structures. The specific surface areas of the samples are calculated by Brunauer−Emmett−Teller (BET). It can be seen that the specific surface areas increase significantly from 72.19 $m^2 g^{-1}$ for $SnO_2$ to 104.01, 131.92 and 139.22 $m^2 g^{-1}$ for $Co_{0.1}Sn_{0.9}O_2$, $Co_{0.2}Sn_{0.8}O_2$ and $Co_{0.3}Sn_{0.7}O_2$, respectively (table 1). The enhancement of specific surface areas of $Co_xSn_{1-x}O_2$ might be attributed to the decreased particles sizes resulting from Co-doped $SnO_2$, as demonstrated by the TEM results. As listed in electronic supplementary material, table S1, the specific surface areas for $40IrO_2/SnO_2$, $40IrO_2/Co_{0.1}Sn_{0.9}O_2$, $40IrO_2/Co_{0.2}Sn_{0.8}O_2$ and $40IrO_2/$

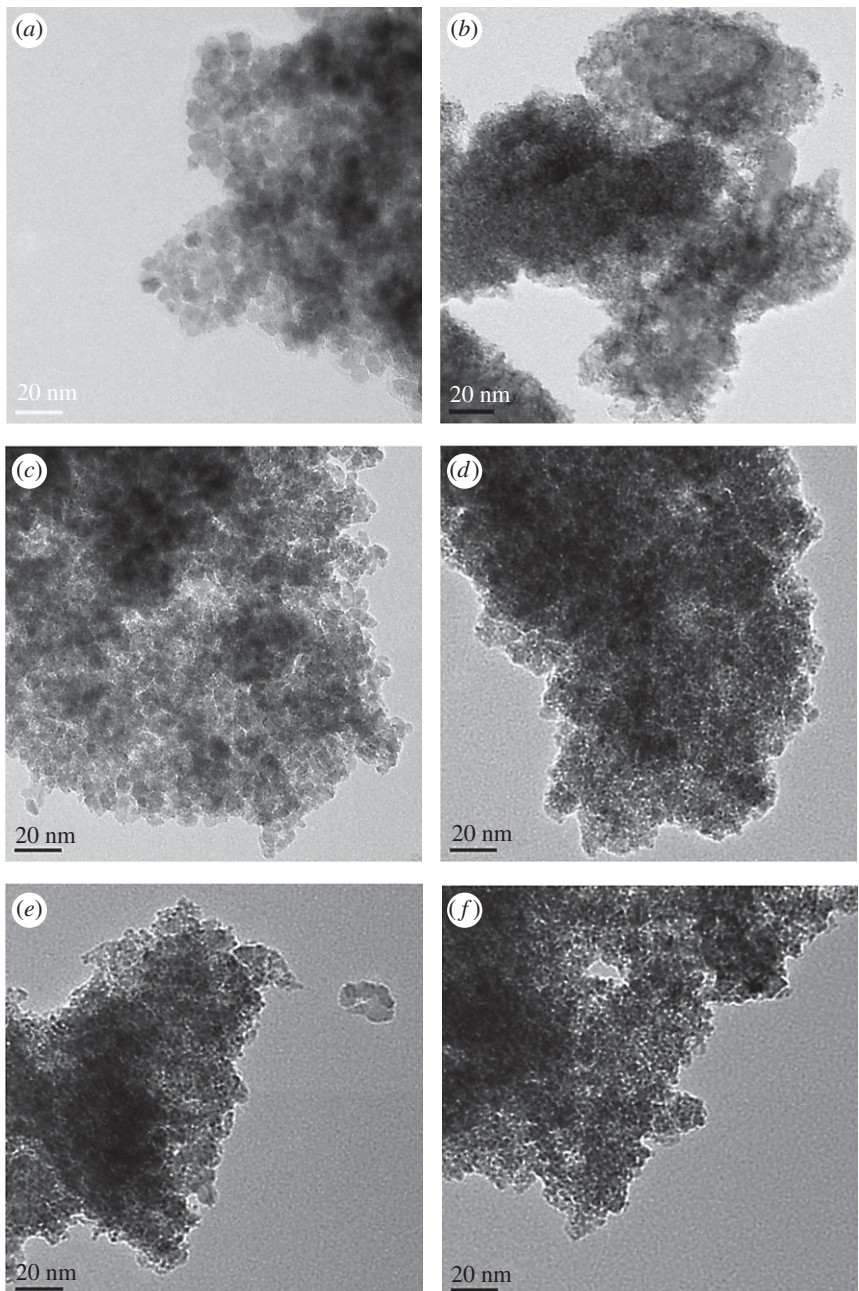

**Figure 2.** TEM images of (*a*) unsupported IrO$_2$, (*b*) SnO$_2$, (*c*) 40IrO$_2$/SnO$_2$ (*d*) 40IrO$_2$/Co$_{0.1}$Sn$_{0.9}$O$_2$, (*e*) 40IrO$_2$/Co$_{0.2}$Sn$_{0.8}$O$_2$ and (*f*) 40IrO$_2$/Co$_{0.3}$Sn$_{0.7}$O$_2$.

Co$_{0.3}$Sn$_{0.7}$O$_2$ are 53.21, 87.54, 91.25, 93.06 m$^2$ g$^{-1}$, respectively. The specific surface areas of Co$_x$Sn$_{1-x}$O$_2$ are higher than those of 40 IrO$_2$/Co$_x$Sn$_{1-x}$O$_2$ ($x$ = 0.1, 0.2, 0.3), which might be due to the pore blocking of Co$_x$Sn$_{1-x}$O$_2$ samples with IrO$_2$ loading. From pore size distributions curves, the pore sizes of Co$_x$Sn$_{1-x}$O$_2$ ($x$ = 0, 0.1, 0.2, 0.3) samples are mainly about 9–10 nm, whereas the mainly pore sizes of 40IrO$_2$/Co$_x$Sn$_{1-x}$O$_2$ ($x$ = 0, 0.1, 0.2, 0.3) samples are mainly about 3–5 nm and 6–7 nm, which could be a result of the IrO$_2$ nanoparticles that occupied a certain amount of pore volume, resulting in a decrease in the pore size [36]. The Co-doped SnO$_2$ samples exhibit a high specific surface area and evident porosity, which might be advantageous for the efficient catalytic performance of the prepared samples.

The XPS spectra were evaluated to reveal the elemental states of each element in the 40IrO$_2$/Co$_{0.2}$Sn$_{0.8}$O$_2$ sample, as shown in figure 5. The high-resolution Ir4$f$ spectrum (figure 5$a$) shows spin–orbit doublets at approximately 61.9 eV and 64.9 eV which can be classified as Ir4$f_{7/2}$ and Ir4$f_{5/2}$, respectively [37]. The binding energy of Ir4$f_{7/2}$ at 61.9 and Ir4$f_{5/2}$ at 64.9 eV is in the form of the Ir$^{4+}$

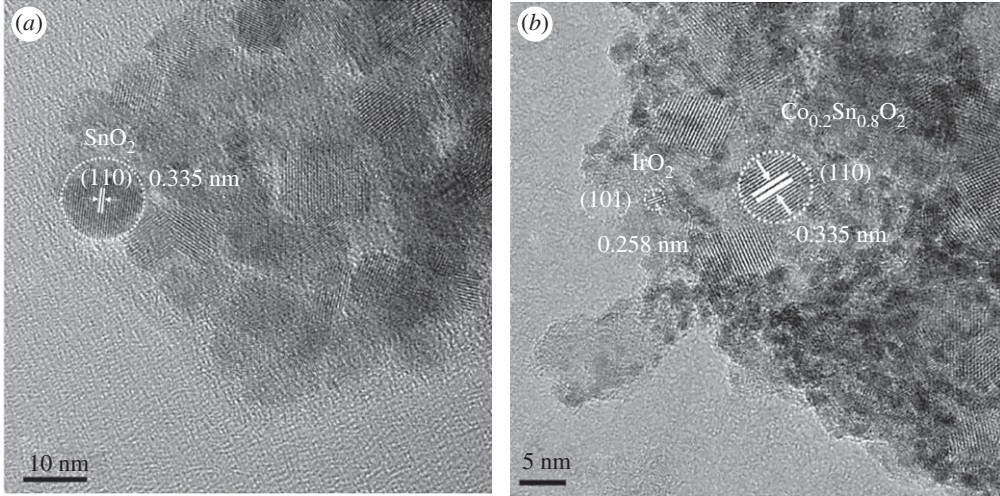

**Figure 3.** HR-TEM images of (*a*) SnO$_2$ and (*b*) 40IrO$_2$/Co$_{0.2}$Sn$_{0.8}$O$_2$.

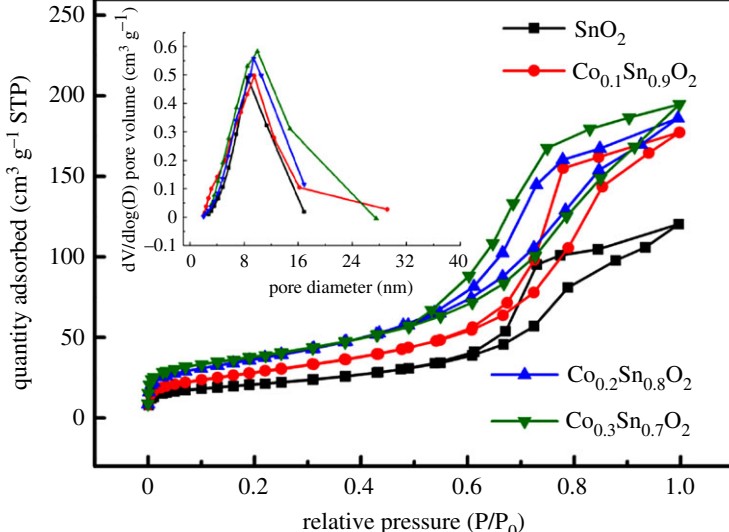

**Figure 4.** N$_2$ adsorption isotherms and pore size distributions (inset) of the Co$_x$Sn$_{1-x}$O$_2$ (*x* = 0, 0.1, 0.2, 0.3) supports.

and Ir4$f_{7/2}$ at 62.9 and Ir4$f_{5/2}$ at 65.9 eV is in accordance with Ir$^{3+}$. Clearly, the atomic ratio of Ir$^{4+}$ is higher than that of Ir$^{3+}$, the ratio of Ir$^{4+}$/Ir$^{3+}$ is 1.42. This indicates that the majority of Ir element in the crystal lattice is Ir$^{4+}$ cations. The Co2$p$ spectrum (figure 5$b$) is deconvoluted into two main components at 781.2 eV for Co2$p_{3/2}$, 797.8 eV for Co2$p_{1/2}$, and two satellite peaks (noted as 'Sat.') are also detected. The presented satellite peaks and the difference of 15.1 eV of the Co2$p_{3/2}$ and Co2$p_{1/2}$ imply that the majority of cobalt is in the states of Co$^{2+}$.[38] figure 5$c$ exhibits the high resolution of Sn3$d$ spectrum. The peaks located at about 487.3 eV and 495.8 eV are the representatives of Sn3$d_{5/2}$ and Sn3$d_{3/2}$, and no other peak is detected, confirming the chemical state of Sn is only in tetravalence [39]. The O 1 s spectrum (figure 4$d$) could be divided into three major peaks: O1 (approx. 529.4 eV), O2 (approx. 531.5 eV) and O3 (approx. 532.9 eV), which are according to the metal-oxygen bonding, oxygen vacancies and hydroxyl species of water molecules adsorbed on the surface, respectively [40].

High electrical conductivity of support is favourable to the supported catalysts for the enhancement of catalytic performance. Prior to the catalytic performance test, the electrical conductivities of all the prepared samples were measured and listed in electronic supplementary material, table S2. The electrical conductivities of SnO$_2$, Co$_{0.1}$Sn$_{0.9}$O$_2$, Co$_{0.2}$Sn$_{0.8}$O$_2$ Co$_{0.3}$Sn$_{0.9}$O$_2$, 40IrO$_2$/SnO$_2$, 40IrO$_2$/Co$_{0.1}$Sn$_{0.9}$O$_2$, 40IrO$_2$/Co$_{0.2}$Sn$_{0.8}$O$_2$, 40IrO$_2$/Co$_{0.3}$Sn$_{0.9}$O$_2$ and unsupported IrO$_2$ are $1.95 \times 10^{-6}$, $2.02 \times 10^{-5}$, $9.51 \times 10^{-5}$, $6.94 \times 10^{-5}$, $6.08 \times 10^{-2}$, $3.15 \times 10^{-1}$, $1.02 \times 10^{0}$, $8.13 \times 10^{-1}$ and $1.02 \times 10^{1}$ S cm$^{-1}$, respectively. The electrical conductivities of Co-doped SnO$_2$ are much higher than those of the pure SnO$_2$, suggesting a favourable effect of the Co dopant on the improvement of the SnO$_2$ conductivity.

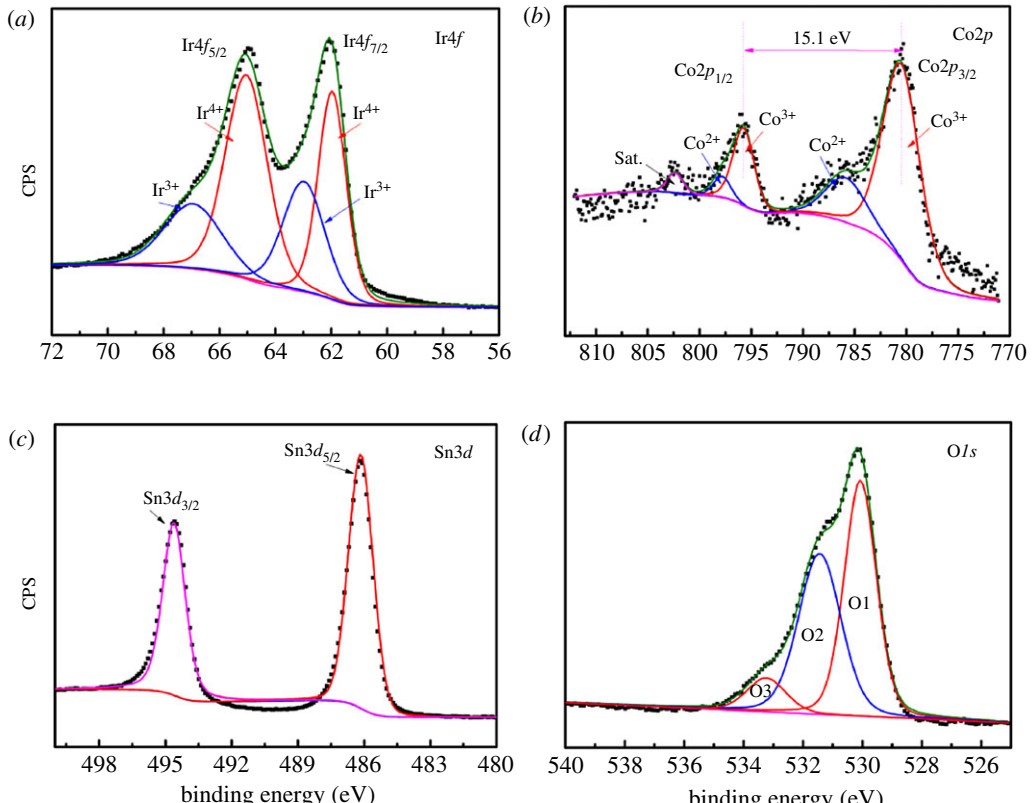

**Figure 5.** The high-resolution XPS spectrum of (a) Ir4f, (b) Co2p, (c) Sn3d and (d) O1s in $40IrO_2/Co_{0.2}Sn_{0.8}O_2$.

After the $IrO_2$ loading, the conductivity of $40IrO_2/Co_xSn_{1-x}O_2$ ($x = 0.1, 0.2, 0.3$) is enhanced because of the excellent electrical conductivity of $IrO_2$. However, the electrical conductivities of $Co_{0.3}Sn_{0.9}O_2$ and $40IrO_2/Co_{0.3}Sn_{0.9}O_2$ are lower than those of $Co_{0.2}Sn_{0.8}O_2$ and $40IrO_2/Co_{0.2}Sn_{0.8}O_2$, which might be due to the increasing impurity scattering centres with the enhancement of Co contents that impede the electron transport, decrease the carrier mobility and reduce electrical conductivity [41].

## 3.2. Electrochemical properties

To evaluate the effect on the surface active sites of catalysts with Co doping, the cyclic voltammograms (CVs) of $40IrO_2/Co_xSn_{1-x}O_2$ were measured in $N_2$-saturated 0.5 M $H_2SO_4$ at a scan rate of 50 mV s$^{-1}$, as shown in figure 6a. For comparison, the CVs of the pristine $SnO_2$ and unsupported $IrO_2$ were also tested at the same condition, and the current densities of all the samples were normalized to the $IrO_2$ loading. The pristine $SnO_2$ shows very low current densities because of the poor electrocatalytic activity. The CVs of $Co_xSn_{1-x}O_2$ samples were also tested (electronic supplementary material, figure S6 (a)) and exhibited higher current densities with Co doping than that of $SnO_2$, but still unsatisfactory. The shapes of the voltammogram of $40IrO_2/Co_xSn_{1-x}O_2$ are similar to those of the unsupported $IrO_2$ but with a higher current density. The CVs of all samples present a broad redox peak of the reversible oxidation and reduction on the $IrO_2$ surface, which suggests a typical pseudo-capacitive behaviour. The voltammetric charge of $40IrO_2/Co_xSn_{1-x}O_2$ and unsupported $IrO_2$, as a function of scan rates, was calculated by the following [42]:

$$Q = \int_{E_0}^{E} \frac{|I|}{v m_{Ir}} \cdot dE,$$   (3.1)

where $I$ is the current density obtained in CV curves, $v = 50$ mV s$^{-1}$ is the scan rate, $m_{Ir}$ is the loading of noble metal Ir on the glassy carbon electrode, $E$ is the scan potential between $-0.148$ and $1.15$ versus Ag/AgCl. The sequence is $40IrO_2/Co_{0.3}Sn_{0.7}O_2$ (333.2 C g(Ir)$^{-1}$) > $40IrO_2/Co_{0.2}Sn_{0.8}O_2$ (314.3 C g(Ir)$^{-1}$) > $40IrO_2/Co_{0.1}Sn_{0.9}O_2$ (252.6 C g(Ir)$^{-1}$) > $40IrO_2/SnO_2$(238.5 C g(Ir)$^{-1}$) > $IrO_2$ (203.5 C g(Ir)$^{-1}$). This confirms the positive effect on the increment of surface active sites with Co doping. The enhancement

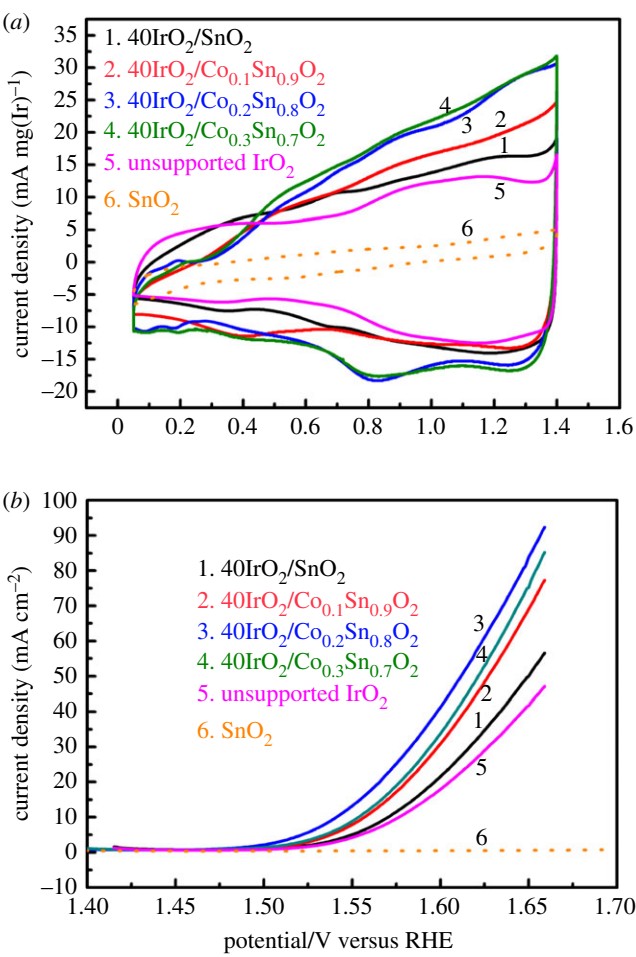

**Figure 6.** (a) Cyclic voltammetry curves of $40IrO_2/Co_xSn_{1-x}O_2$, pristine $SnO_2$ and unsupported $IrO_2$ in $N_2$-saturated 0.5 M $H_2SO_4$ solution at a scan rate is 50 mV s$^{-1}$ and (b) LSV curves of $40IrO_2/Co_xSn_{1-x}O_2$, pristine $SnO_2$ and unsupported $IrO_2$.

**Table 2.** The obtained values of voltammetric charge (C/g), the potentials at 10 mA cm$^{-2}$, cell potential at 1 A cm$^{-2}$, $R_\Omega$ and $R_{ct}$ of the prepared samples.

| samples | voltammetric charge (C g$^{-1}$) | LSV | | EIS (mΩ cm$^2$) | |
| --- | --- | --- | --- | --- | --- |
| | | The potentials at 10 mA cm$^{-2}$ | cell potential at 1 A cm$^{-2}$ | $R_\Omega$ | $R_{ct}$ |
| $40IrO_2/SnO_2$ | 238.5 | 1.570 | 1.847 | 152.0 | 40.3 |
| $40IrO_2/Co_{0.1}Sn_{0.9}O_2$ | 252.6 | 1.557 | 1.812 | 114.2 | 37.0 |
| $40IrO_2/Co_{0.2}Sn_{0.8}O_2$ | 314.3 | 1.541 | 1.748 | 74.7 | 32.1 |
| $40IrO_2/Co_{0.3}Sn_{0.7}O_2$ | 333.2 | 1.554 | 1.77 | 85.3 | 34.3 |
| unsupported $IrO_2$ | 203.5 | 1.577 | 1.713 | 63.1 | 52.6 |

of surface active sites could be ascribed to the decreased particle sizes of $SnO_2$ supports that provide more sites for $IrO_2$ dispersion. Consequently, according to the calculated voltammetric charge, it is expected that the OER catalytic activity of the prepared samples would be improved as the Co-doped samples. The overall OER performance of the samples will be discussed in the following single cell test results. Figure 6b shows the LSV polarization curves of the pristine $SnO_2$, unsupported $IrO_2$ and $40IrO_2/Co_xSn_{1-x}O_2$ samples in $N_2$-saturated 0.5 M $H_2SO_4$ at a scan rate of 50 mV s$^{-1}$. The potentials at the current density of 10 mA cm$^{-2}$ are listed in table 2. The measured potentials are 1.577, 1.570, 1.557, 1.541 and 1.554 V versus RHE for unsupported $IrO_2$, $40IrO_2/SnO_2$, $40IrO_2/Co_{0.1}Sn_{0.9}O_2$,

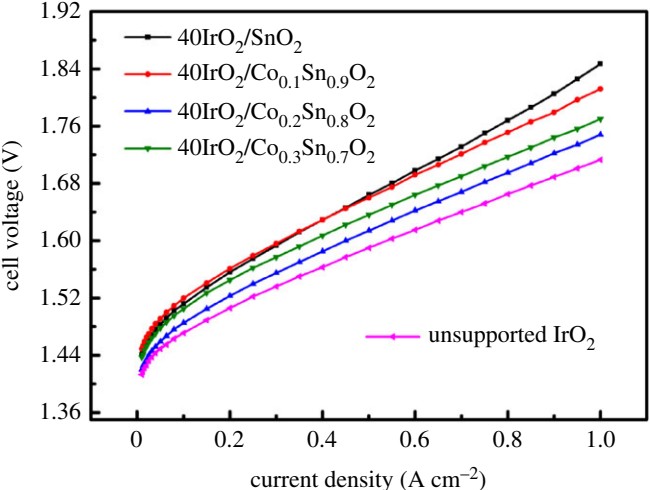

**Figure 7.** Polarization curves of single cells equipped with 40IrO$_2$/Co$_x$Sn$_{1-x}$O$_2$ ($x$ = 0, 0.1, 0.2, 0.3) and unsupported IrO$_2$ at 80°C.

40IrO$_2$/Co$_{0.2}$Sn$_{0.8}$O$_2$ and 40IrO$_2$/Co$_{0.3}$Sn$_{0.7}$O$_2$, respectively. It is clearly observed that 40IrO$_2$/Co$_{0.2}$Sn$_{0.8}$O$_2$ reveals the lowest overpotential on mass activity, indicating Co-doped support could favour the increment of the active substance and enhance the OER performance. Electronic supplementary material, figure S6 (b) displays the LSV polarization curves of the Co$_x$Sn$_{1-x}$O$_2$ samples. It is clearly observed that the overpotential of Co$_x$Sn$_{1-x}$O$_2$ samples is higher than that of 40IrO$_2$/Co$_x$Sn$_{1-x}$O$_2$ samples, indicating the low catalytic activity of supports.

## 3.3. Electrolysis cell performance

After the MEAs assembled in single cells, the OER performance of the 40IrO$_2$/Co$_x$Sn$_{1-x}$O$_2$ catalysts was further characterized by I-V polarization measurement from 0.01 to 1 A cm$^{-2}$ at 80°C, as displayed in figure 7. Again, the OER performance of unsupported IrO$_2$ was detected at the same condition for comparison. For the low current density (less than 0.1 A cm$^{-2}$), the cell voltage of the 40IrO$_2$/Co$_x$Sn$_{1-x}$O$_2$ increases and unsupported IrO$_2$ increases nonlinearly along the current density, which is mainly affected by activation polarization. Once the polarization current density increases gradually, the disparity in the OER performance is mainly due to ohmic resistance and the polarization resistance, as presented by the linear increase of potential for all samples. Compared to the 40IrO$_2$/SnO$_2$, the cell voltage of 40IrO$_2$/Co$_x$Sn$_{1-x}$O$_2$ shows a superior performance. The 40IrO$_2$/Co$_{0.2}$Sn$_{0.8}$O$_2$ exhibits optimal activity at 1A cm$^{-2}$, followed by the order of 40IrO$_2$/Co$_{0.3}$Sn$_{0.7}$O$_2$, 40IrO$_2$/Co$_{0.1}$Sn$_{0.9}$O$_2$, 40IrO$_2$/SnO$_2$, in that order. However, the OER performance of the 40IrO$_2$/Co$_x$Sn$_{1-x}$O$_2$ catalysts is still lower than that of unsupported IrO$_2$ due to the lower amount of Ir loading. The sequence in OER performance at 1 A cm$^{-2}$ is IrO$_2$ (1.713 V) < 40IrO$_2$/Co$_{0.2}$Sn$_{0.8}$O$_2$ (1.748 V) < 40IrO$_2$/Co$_{0.3}$Sn$_{0.7}$O$_2$ (1.770 V) < 40IrO$_2$/Co$_{0.1}$Sn$_{0.9}$O$_2$ (1.812 V) < 40IrO$_2$/SnO$_2$ (1.847 V). Notably, the OER performance decreases when the Co-doping level reaches to the point $x$ = 0.3. This could be attributed to the increasing impurity scattering centres with the enhancement of Co contents that impede the electron transport and decrease the carrier mobility, leading to the degradation of OER performance [41].

The charge transfer resistance (R$_{ct}$) is a critically important parameter in reflection of reaction kinetics in electrocatalytic performance for a catalyst, and a lower R$_{ct}$ implies a faster reaction rate [43]. To further reveal the intrinsic charge transfer properties of IrO$_2$ supported by varying Co-doped SnO$_2$ as supports, the EIS measurements were conducted at 0.3 A cm$^{-2}$ in the single cells. Figure 8 exhibits the Nyquist plots of 40IrO$_2$/Co$_x$Sn$_{1-x}$O$_2$ and unsupported IrO$_2$, and the appropriate equivalent circuit model is shown in the inset. R$_{ct}$ is the charge transfer resistance of a faradic process occurring at the interface of catalyst and electrolyte, which is evaluated from the large semicircles by the difference between low- and high-frequency intercepts on the real axis [44]. The ohmic resistance ($R_\Omega$) is the series resistance of all the geometry compositions in a cell. $R_\Omega$ can be calculated from the intercept extending from the high-frequency side of the curve of the real axis [45]. The calculated values of the R$_{ct}$ and $R_\Omega$ are listed in table 2. It can be seen that the unsupported IrO$_2$ shows a lower $R_\Omega$ (63.1 m$\Omega$ cm$^2$) because of the excellent electrical conductivity than 40IrO$_2$/Co$_x$Sn$_{1-x}$O$_2$. The Rct value of 40IrO$_2$/Co$_{0.2}$Sn$_{0.8}$O$_2$ is 32 m$\Omega$ cm$^2$ and exhibits the lowest charge transfer resistance, followed by

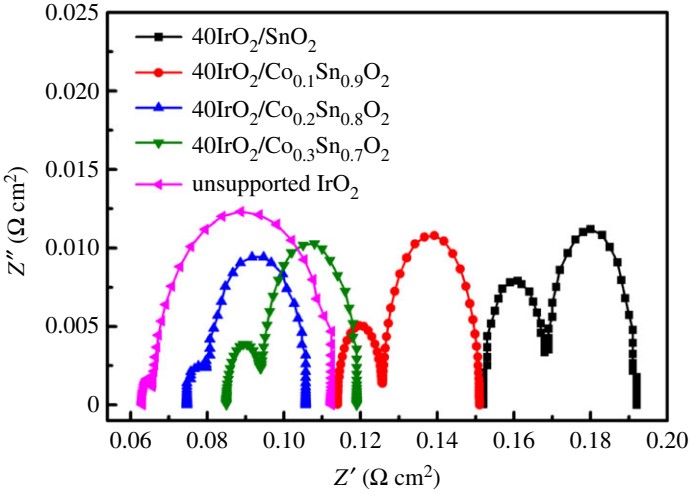

**Figure 8.** Nyquist diagrams of $40IrO_2/Co_xSn_{1-x}O_2$ ($x = 0$, 0.1, 0.2, 0.3) and unsupported $IrO_2$ at 0.3 A cm$^{-2}$ and 80°C.

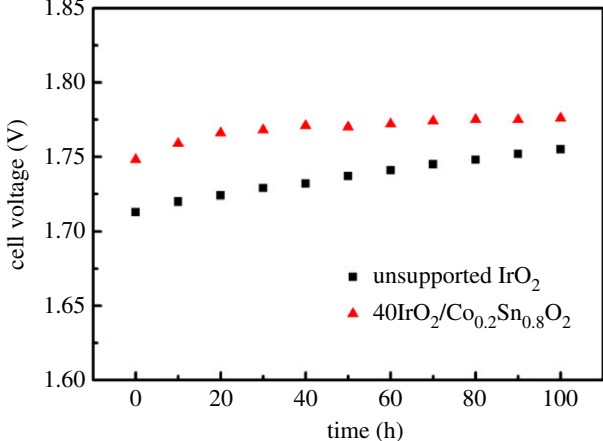

**Figure 9.** Durability test of the unsupported $IrO_2$ and $40IrO_2/Co_{0.2}Sn_{0.8}O_2$ sample in a single cell at 1 A cm$^{-2}$ at 80°C.

$40IrO_2/Co_{0.3}Sn_{0.7}O_2$ (34 mΩ cm$^2$), $40IrO_2/Co_{0.1}Sn_{0.9}O_2$ (37 mΩ cm$^2$), $40IrO_2/SnO_2$ (40 mΩ cm$^2$) and unsupported $IrO_2$ (52 mΩ cm$^2$) in sequence. It is anticipated that the significant increase of the exposed surface area of $IrO_2$ could enhance surface active sites and charge transport property, leading to the improvement of OER performance of $40IrO_2/Co_xSn_{1-x}O_2$.

Long-term durability is a critical parameter for the commercialization of catalysts. The durability test of the unsupported $IrO_2$ and $40IrO_2/Co_{0.2}Sn_{0.8}O_2$ catalysts in a single cell was measured at current densities of 1 A cm$^{-2}$ at 80°C for 100 h. As displayed in figure 9, both the cell potentials of the samples reveal different degrees of increase. The cell potential of the unsupported $IrO_2$ shows a relatively uniform upward trend, rising from 1.713 to 1.755 V after 100 h, and the average decay is 0.42 mV h$^{-1}$. The cell voltage of $40IrO_2/Co_{0.2}Sn_{0.8}O_2$ increases slightly during the initial 40 h, but later remains almost constant at 1.766 V, and the average degradation rate is 0.18 mV h$^{-1}$. According to Kötz [46], the alternation between Ir(III) and Ir(IV) during the catalysis of OER plays a crucial role in the oxidation of the hydroxyl, while Ir(VI) is easily prone to corrosion according to the following [47]:

$$IrO_2 + H_2O \rightarrow IrO_4^{2-} + 2H^+. \tag{3.2}$$

This implies that the supports are beneficial in the formation of stable Ir oxide during the modified Adams fusion, and that Co-doped content might further enhance the durability of $40IrO_2/Co_xSn_{(1-x)}O_2$ catalyst. Additionally, recent studies on OER performance and stability of iridium-based catalysts have been summarized and listed in table 3. It can be seen that the cell voltage and degradation rate of $40IrO_2/Co_{0.2}Sn_{0.8}O_2$ are comparable to or superior to those of the previously reported iridium-based catalysts (such as $IrO_2$/V-doped $TiO_2$ [31] and $IrO_2$-ATO [48]). This further

**Table 3.** OER performance and stability reported in the literature for iridium-based catalysts.

| references | anode catalyst | Ir loading (mg cm$^{-2}$) | cathode catalyst | Pt loading (mg cm$^{-2}$) | operating current (A cm$^{-2}$) | operating temperature (°C) | cell voltage | electrode fabrication process | active area (cm$^2$) | test time (h) | degradation rate (μV h$^{-1}$) |
|---|---|---|---|---|---|---|---|---|---|---|---|
| Hao et al. [31] | IrO$_2$/V -doped TiO$_2$ | 2.5 | Pt/C | 0.5 | 1 | 80 | 2.015 V (1 A cm$^{-2}$) | spraying | 3.65 | 50 | 1980 |
| Puthiyapura et al. [48] | IrO$_2$-ATO | 2 | Pt/C | 0.5 | 1.0 | 80 | 1.80 V (1 A cm$^{-2}$) | spraying | 1 | | |
| Rakousky et al. [49] | IrO$_2$ and TiO$_2$ | 2.25 | Pt/C | 0.8 | 2.0 | 80 | 1.84 V (2 A cm$^{-2}$) | commercial | 17.64 | 1150 | 194 |
| Zeng et al. [50] | Ir black | 0.873 | Pt/C | 1.0 | 0.25 | 80 | 1.728 V (2 A cm$^{-2}$) | spraying | 4 | 300 | 52 |
| Faustini et al. [51] | Ir$_{0.7}$Ru$_{0.3}$O$_x$ | 1.8 | Pt/C | 0.5 | 1 | 80 | 1.680 V (1 A cm$^{-2}$) | spraying | 6.25 | | |
| Jorge et al. [52] | gCNH-IrO2 | 1.2 | Pt/C | 4 | 1 | 80 | 1.93 V (1 A cm$^{-2}$) | spraying | 7.07 | | |
| this study | IrO$_2$/Co$_{0.2}$Sn$_{0.8}$O$_2$ | 2.5 | Pt/C | 0.5 | 1 | 80 | 1.776 V (1 A cm$^{-2}$) | spraying | 3.65 | 100 | 180 |

confirms that the prepared Co-doped $SnO_2$ as a support for $IrO_2$ catalysts is a potential candidate for the practical application of SPE water electrolyzer.

# 4. Conclusion

A varying amount of Co-doped $SnO_2$ as anode support for $IrO_2$ has been successfully prepared and then characterized by series methods. The particle sizes of the prepared $Co_xSn_{(1-x)}O_2$ ($x = 0.1, 0.2, 0.3$) samples were decreased with the increase of Co-doping content, which provided more sites for the well-dispersed $IrO_2$. Also, the prepared samples exhibited increased high specific surface areas. The $40IrO_2/Co_{0.2}Sn_{0.8}O_2$ exhibited the lowest overpotential with a cell potential of 1.748 V at 1000 mA cm$^{-2}$ and showed a good stability during the 100 h operating at the current density 1 A cm$^{-2}$ at 80°C. The decreased overpotential of $40IrO_2/Co_{0.2}Sn_{0.8}O_2$ could be ascribed to the increment of surface active sites, the enhancement of electrical conductivity and the higher charge transfer, as verified by CVs and EIS measurement results. Consequently, the $Co_xSn_{(1-x)}O_2$ shows a promising alternative support for anode catalysts in the SPE water electrolyzer.

Data accessibility. The datasets supporting this article have been uploaded as the electronic supplementary material.

Authors' contributions. All authors made substantial contributions to this paper. G.C. and H.L. supervised and directed the project and conceived and designed the experiments. G.C., J.K.L. and S.W. analysed the data and wrote the manuscript. All authors commented on and revised the paper at multiple stages and approved the final version for publication.

Competing interests. We declare we have no competing interests.

Funding. This study was supported by the National Nature Science Foundation of China (no. 21306141), the National High-tech R&D Program of China (no. 2012AA053301) and the Fundamental Research Funds for the Central Universities (no. kx0170920173391).

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
