## [Reviewer comments · Royal Society Open Science]

Review History

RSOS-182223.R0 (Original submission)

Review form: Reviewer 1

Is the manuscript scientifically sound in its present form?

Yes

Are the interpretations and conclusions justified by the results?

Yes

Is the language acceptable?

Yes

Is it clear how to access all supporting data?

Yes

Do you have any ethical concerns with this paper?

No

Have you any concerns about statistical analyses in this paper?

No

Recommendation?

Accept with minor revision (please list in comments)

Comments to the Author(s)

In this study, the author systemically investigated the doping effect of cobalt into the SnO₂ crystal, which serves as excellent support for noble metal water oxidation catalysts but usually encounters poor electronic conductivity. The influence of cobalt doping on the SnO₂ support including the grain size, morphology, and also catalytic performance was studied in detail based on the solid characterizations. The manuscript was well-designed and main assumption could be certified by the characterizations, which makes it very suitable for publication in this journal only after addressing the minor concern of the reviewer as listed below:

1. The doping of hetero-atoms into the crystal structure of material could also bring in undesired structural defects, which would in some case function as charge recombination center. Some discussion on this concern should be considered.
2. The author discussed the promising role of hydrogen fuel for the development of novel clean energy and relieving the global environment issues. Thus, it is reasonable to concise include the new techniques of hydrogen production developed in the community, such as electrochemical hydrogen production, photocatalytic/photo-electrochemical water splitting: Nat. Mater., 2006, 5, 909-913; Angew. Chem. Int. Ed. 2019 DOI: 10.1002/anie.201811938; Nat. Mater. 2017, 16, 646, J. Mater. Chem. A, 2017, 5, 12723-12728.
3. It would be good supplementary and valuable information for the general readers if the author could add the comparison of the catalytic performance of the current study with the previous system.

Review form: Reviewer 2

Is the manuscript scientifically sound in its present form?

Yes

Are the interpretations and conclusions justified by the results?

Yes

Is the language acceptable?

Yes

Is it clear how to access all supporting data?

Yes

Do you have any ethical concerns with this paper?

No

Have you any concerns about statistical analyses in this paper?

No

Recommendation?

Accept with minor revision (please list in comments)

Comments to the Author(s)

This manuscript reports a novel $40\text{IrO}_2/\text{Co}_x\text{Sn}(1-x)\text{O}_2$ ($X = 0.1, 0.2, 0.3$) anode catalyst by a facile method and their application in solid polymer electrolyte (SPE) water electrolyzer. Impressively, the $40\text{IrO}_2/\text{Co}_{0.2}\text{Sn}_{0.8}\text{O}_2$ exhibits the lowest overpotential (1.748 V at 1000 mA cm^{-2}), and only 0.18 mV h⁻¹ of voltage increased for 100 h durability test at 1000 mA cm^{-2} . Many details are adequately handled and appropriate techniques have been used. But there are many grammatical errors and messy layout. Therefore, this paper can be published in R. Soc. open sci. after minor revision on the following points.

1. What is the state of cobalt? Elemental or compound or both? In XRD description, "This confirms that Co was successfully doped into SnO_2 ." In XPS description, "The presented satellite peaks and the difference of 15.1 eV of the $\text{Co}2p_{3/2}$ and $\text{Co}2p_{1/2}$ imply that the majority of cobalt is in the states of Co^{2+} ." The authors should describe it properly.
2. The specific surface areas and pore size distributions of the $40\text{IrO}_2/\text{Co}_x\text{Sn}(1-x)\text{O}_2$ samples should be provided.
3. "It can be seen that the unsupported IrO_2 shows a lower R_Ω ($76 \text{ m}\Omega \text{ cm}^2$) because of the excellent electrical conductivity than $40\text{IrO}_2/\text{Co}_{0.2}\text{Sn}_{0.8}\text{O}_2$." in page 6 of line 8, is there any reference of the relationship between the ohmic resistance and electrical conductivity? If not, you should determine the electrical conductivity of IrO_2 and $40\text{IrO}_2/\text{Co}_{0.2}\text{Sn}_{0.8}\text{O}_2$.
4. Fig.3 and Fig. 6 captions need be checked carefully. It is (a) and (b) rather than (f) and (g) in Fig.3 caption. In Fig. 6, (c) and (d) picture cannot be founded.
5. There are many messy layout and grammatical errors. For example, page 2, line 41, space is missing between number and unit; page 3, line 12, "ml" should be "mL".

Decision letter (RSOS-182223.R0)

25-Feb-2019

Dear Professor Chen:

Title: Mesoporous $\text{Co}_x\text{Sn}(1-x)\text{O}_2$ as an efficient oxygen evolution catalyst support for SPE water electrolyzer
 Manuscript ID: RSOS-182223

Thank you for submitting the above manuscript to Royal Society Open Science. On behalf of the Editors and the Royal Society of Chemistry, I am pleased to inform you that your manuscript will be accepted for publication in Royal Society Open Science subject to minor revision in accordance with the referee suggestions. Please find the reviewers' comments at the end of this email.

The reviewers and handling editors have recommended publication, but also suggest some minor revisions to your manuscript. Therefore, I invite you to respond to the comments and revise your manuscript.

Please also include the following statements alongside the other end statements. As we cannot publish your manuscript without these end statements included, if you feel that a given heading is not relevant to your paper, please nevertheless include the heading and explicitly state that it is not relevant to your work. We have included a screenshot example of the end statements for reference.

- Acknowledgements

Because the schedule for publication is very tight, it is a condition of publication that you submit the revised version of your manuscript before 06-Mar-2019. Please note that the revision deadline will expire at 00.00am on this date. If you do not think you will be able to meet this date please let me know immediately.

Best wishes,
Dr Laura Smith
Publishing Editor, Journals

On behalf of the Subject Editor Professor Anthony Stace and the Associate Editor Professor Claire Carmalt.

RSC Associate Editor:
Comments to the Author:
(There are no comments.)

RSC Subject Editor:
Comments to the Author:
(There are no comments.)

Reviewer comments to Author:
Reviewer: 1

Comments to the Author(s)

In this study, the author systemically investigated the doping effect of cobalt into the SnO₂ crystal, which serves as excellent support for noble metal water oxidation catalysts but usually encounters poor electronic conductivity. The influence of cobalt doping on the SnO₂ support including the grain size, morphology, and also catalytic performance was studied in detail based on the solid characterizations. The manuscript was well-designed and main assumption could be certified by the characterizations, which makes it very suitable for publication in this journal only after addressing the minor concern of the reviewer as listed below:

1. The doping of hetero-atoms into the crystal structure of material could also bring in undesired structural defects, which would in some case function as charge recombination center. Some discussion on this concern should be considered.
2. The author discussed the promising role of hydrogen fuel for the development of novel clean energy and relieving the global environment issues. Thus, it is reasonable to concise include the new techniques of hydrogen production developed in the community, such as electrochemical hydrogen production, photocatalytic/photo-electrochemical water splitting: *Nat. Mater.*, 2006, 5, 909-913; *Angew. Chem. Int. Ed.* 2019 DOI: 10.1002/anie.201811938; *Nat. Mater.* 2017, 16, 646, *J. Mater. Chem. A*, 2017, 5, 12723-12728.
3. It would be good supplementary and valuable information for the general readers if the author could add the comparison of the catalytic performance of the current study with the previous system.

Reviewer: 2

Comments to the Author(s)

This manuscript reports a novel 40IrO₂/CoxSn(1-x)O₂ (X= 0.1, 0.2, 0.3) anode catalyst by a facile method and their application in solid polymer electrolyte (SPE) water electrolyzer. Impressively, the 40IrO₂/Co_{0.2}Sn_{0.8}O₂ exhibits the lowest overpotential (1.748 V at 1000 mA cm⁻²), and only

0.18 mV h⁻¹ of voltage increased for 100 h durability test at 1000 mA cm⁻². Many details are adequately handled and appropriate techniques have been used. But there are many grammatical errors and messy layout. Therefore, this paper can be published in R. Soc. open sci. after minor revision on the following points.

1. What is the state of cobalt? Elemental or compound or both? In XRD description, "This confirms that Co was successfully doped into SnO₂." In XPS description, "The presented satellite peaks and the difference of 15.1 eV of the Co2p_{3/2} and Co2p_{1/2} imply that the majority of cobalt is in the states of Co²⁺." The authors should describe it properly.
2. The specific surface areas and pore size distributions of the 40IrO₂/Co_xSn_(1-x)O₂ samples should be provided.
3. "It can be seen that the unsupported IrO₂ shows a lower R_Ω (76 mΩ cm²) because of the excellent electrical conductivity than 40IrO₂/Co_{0.2}Sn_{0.8}O₂." in page 6 of line 8, is there any reference of the relationship between the ohmic resistance and electrical conductivity? If not, you should determine the electrical conductivity of IrO₂ and 40IrO₂/Co_{0.2}Sn_{0.8}O₂.
4. Fig.3 and Fig. 6 captions need be checked carefully. It is (a) and (b) rather than (f) and (g) in Fig.3 caption. In Fig. 6, (c) and (d) picture cannot be founded.
5. There are many messy layout and grammatical errors. For example, page 2, line 41, space is missing between number and unit; page 3, line 12, "ml" should be "mL".

Author's Response to Decision Letter for (RSOS-182223.R0)

See Appendix A.

Decision letter (RSOS-182223.R1)

26-Mar-2019

Dear Professor Chen:

Title: Mesoporous Co_xSn_(1-x)O₂ as an efficient oxygen evolution catalyst support for SPE water electrolyzer

Manuscript ID: RSOS-182223.R1

It is a pleasure to accept your manuscript in its current form for publication in Royal Society Open Science. The chemistry content of Royal Society Open Science is published in collaboration with the Royal Society of Chemistry.

On behalf of the Subject Editor Professor Anthony Stace and the Associate Editor Professor Claire Carmalt.

RSC Associate Editor
Comments to the Author:
The manuscript has been corrected appropriately and accept as is recommended.

Reviewer(s)' Comments to Author:

Appendix A

Prof. Dr. Gang Chen

College of Materials Science and Engineering, Hunan University, Changsha 410012, Hunan, China

E-mail: chengang@hnu.edu.cn

March 4th, 2018

Dear editor,

Thank you and reviewers very much for the comments on this manuscript (**Manuscript No.: RSOS-182223**) entitled “*Mesoporous $\text{Co}_x\text{Sn}_{(1-x)}\text{O}_2$ as an efficient oxygen evolution catalyst support for SPE water electrolyzer*” by Gang Chen*, Jiankun Li, Hong Lv*, Sen Wang, Jian Zuo, Lihua Zhu. According to the comments, we made some necessary amendments in the revised version. The replies to the reviewers’ questions are listed as follows. We hope the revised version is entirely suitable for publication in *Royal Society Open Science*.

Replies to reviewers’ comments

Reviewer: 1

Comments:

In this study, the author systemically investigated the doping effect of cobalt into the SnO_2 crystal, which serves as excellent support for noble metal water oxidation catalysts but usually encounters poor electronic conductivity. The influence of cobalt doping on the SnO_2 support including the grain size, morphology, and also catalytic performance was studied in detail based on the solid characterizations. The manuscript was well-designed and main assumption could be certified by the characterizations, which makes it very suitable for publication in this journal only after addressing the minor concern of the reviewer as listed below:

To Reviewer #1:

Q1: The doping of hetero-atoms into the crystal structure of material could also bring in undesired structural defects, which would in some case function as charge recombination center. Some discussion on this concern should be considered.

Reply: Thanks a lot for the reviewer’s suggestion. According to the suggestions, some discussion on this concern was added in the revised manuscript. (*Please see page 8, line 2 in the revised*

manuscript: ...Notably, the OER performance decrease when the Co doping level arrives $x=0.3$. This could be attributed to the increasing impurity scattering centers with the enhancement of Co contents impede the electron transport and decrease the carrier mobility, leading the degradation of OER performance. [43])

Q2: The author discussed the promising role of hydrogen fuel for the development of novel clean energy and relieving the global environment issues. Thus, it is reasonable to concise include the new techniques of hydrogen production developed in the community, such as electrochemical hydrogen production, photocatalytic/photo-electrochemical water splitting: Nat. Mater., 2006, 5, 909-913; Angew. Chem. Int. Ed. 2019 DOI: 10.1002/anie.201811938; Nat. Mater. 2017, 16, 646, J. Mater. Chem. A, 2017, 5, 12723-12728.

Reply: Thanks a lot for the reviewer's suggestion. According to the suggestions, the new techniques of hydrogen production developed in the community was added in the revised manuscript. (*Please see page 1, part of Introduction in the revised manuscript: Hydrogen is regarded ...present on Earth.[1] Water splitting for hydrogen generation is a major component of modern clean energy technologies,[2] such as water-alkali electrolyzers,[3] solid polymer electrolyte (SPE) water electrolyzers[4] and photocatalytic/photo-electrochemical water splitting.[5-8] Among these techniques,...)*)

Q3: It would be good supplementary and valuable information for the general readers if the author could add the comparison of the catalytic performance of the current study with the previous system.

Reply: Thanks a lot for the reviewer's suggestion. This is very good suggestion. According to the suggestion, the comparison of the catalytic performance of the current study with the previous system was added in the revised manuscript. (*Please see page 8, line 36 in the revised manuscript and the OER performance and stability of iridium-based catalysts reported in the previous literature listed in Table 3: Additionally, recent studies on OER performance and stability of iridium-based catalysts have been summarized and listed in Table 3. It can be seen that the cell voltage and degradation rate of $40\text{IrO}_2/\text{Co}_{0.2}\text{Sn}_{0.8}\text{O}_2$ is comparable to or superior to those of the previously reported iridium--based catalysts (such as $\text{IrO}_2/\text{V-doped TiO}_2$ [32] and $\text{IrO}_2\text{-ATO}$ [50]). This further indicates that the prepared Co doped SnO_2 as supports for IrO_2 catalysts is a potential candidate for the practical application of SPE water electrolyzer.)*)

Reviewer: 2

Comments:

This manuscript reports a novel $40\text{IrO}_2/\text{Co}_x\text{Sn}_{(1-x)}\text{O}_2$ ($X=0.1, 0.2, 0.3$) anode catalyst by a facile method and their application in solid polymer electrolyte (SPE) water electrolyzer. Impressively, the $40\text{IrO}_2/\text{Co}_{0.2}\text{Sn}_{0.8}\text{O}_2$ exhibits the lowest overpotential (1.748 V at 1000 mA cm^{-2}), and only 0.18 mV h⁻¹ of voltage increased for 100 h durability test at 1000 mA cm^{-2} . Many details are adequately handled and appropriate techniques have been used. But there are many grammatical errors and messy layout. Therefore, this paper can be published in R. Soc. open sci. after minor revision on the following points.

To Reviewer #2:

Q1: What is the state of cobalt? Elemental or compound or both? In XRD description, “This confirms that Co was successfully doped into SnO_2 .” In XPS description, “The presented satellite peaks and the difference of 15.1 eV of the $\text{Co}2p_{3/2}$ and $\text{Co}2p_{1/2}$ imply that the majority of cobalt is in the states of Co^{2+} .” The authors should describe it properly.

Reply: Thanks a lot for the reviewer’s suggestion. We apologize that our previous claim was not accurate. In the XPS characterization analysis, the valence state of cobalt was determined as bivalence, which is also similar with those reported previously (*e.g.*, Journal of Materials Chemistry A. 2018, **6**, 7592-7607; ACS Appl. Mater. Inter. 2016, **8**, 28917). So, the description of “This confirms that Co was successfully doped into SnO_2 .” was misleading statements. We have revised the related statement in the revised manuscript. (*Please see page 4, part of 4.1, line 6-7 in the revised manuscript: ...which confirms that Co ions were successfully doped into SnO_2 . [34]...*)

Q2: The specific surface areas and pore size distributions of the $40\text{IrO}_2/\text{Co}_x\text{Sn}_{(1-x)}\text{O}_2$ samples should be provided.

Reply: Thanks a lot for the reviewer’s suggestion. According to the suggestions, we have added the specific surface areas and pore size distributions of the $40\text{IrO}_2/\text{Co}_x\text{Sn}_{(1-x)}\text{O}_2$ ($x=0, 0.1, 0.2, 0.3$) samples in the revised Supporting Information. (*Please see the revised manuscript, Page 5, paragraph 3: Fig. 4, Fig. S7, and Table S1. It is well known that the specific ... measurements were carried out. Fig. 4 and Fig. S7 represent the N_2 adsorption-desorption isotherms of the $\text{Co}_x\text{Sn}_{1-x}\text{O}_2$ and $40\text{IrO}_2/\text{Co}_x\text{Sn}_{1-x}\text{O}_2$ ($x=0, 0.1, 0.2, 0.3$), the inset are the corresponding Barrett–Joyner–Halenda (BJH) pore size distribution curves. Typical Langmuir type IV with an inherent hysteresis loop ... , as demonstrated by TEM results. As listed in Table S1, the specific*

surface areas for 40IrO₂/SnO₂, 40IrO₂/Co_{0.1}Sn_{0.9}O₂, 40IrO₂/Co_{0.2}Sn_{0.8}O₂ and 40IrO₂/Co_{0.3}Sn_{0.7}O₂ are 53.21, 87.54, 91.25, 93.06 m²·g⁻¹, respectively. The specific surface area of Co_xSn_{1-x}O₂ are higher than that of 40 IrO₂/Co_xSn_{1-x}O₂ (x= 0.1, 0.2, 0.3), which might be due to the pore blocking of Co_xSn_{1-x}O₂ samples with IrO₂ loading. From pore size distributions curves, Co_xSn_{1-x}O₂ (x= 0, 0.1, 0.2, 0.3) samples exhibit the mainly pore sizes are about 9-10 nm, whereas the mainly pore sizes of 40IrO₂/Co_xSn_{1-x}O₂ (x= 0, 0.1, 0.2, 0.3) samples located at about 3-5 nm and 6-7 nm, which could be resulted from the IrO₂ nanoparticles occupied a certain amount of pore volume, resulting in a decrease in the pore size.[38]...)

Q3: It can be seen that the unsupported IrO₂ shows a lower R_Ω (76 mΩ cm²) because of the excellent electrical conductivity than 40IrO₂/Co_{0.2}Sn_{0.8}O₂.” in page 6 of line 8, is there any reference of the relationship between the ohmic resistance and electrical conductivity? If not, you should determine the electrical conductivity of IrO₂ and 40IrO₂/Co_{0.2}Sn_{0.8}O₂.

Reply: Thanks a lot for the reviewer’s suggestion. The R_Ω is the whole ohmic resistance of SPE water electrolyzer including Nafion 117 membrane, catalyst layer, bipolar plate, Ti mesh, carbon paper and wire. The assembled single cells differ only in their anode catalytic layers, which means the difference of R_Ω comes from the anode catalysts. The electrical conductivity experiments of supports Co_xSn_{1-x}O₂, 40IrO₂/Co_xSn_{1-x}O₂ (x= 0, 0.1, 0.2, 0.3) and unsupported IrO₂ were also performed. The measured values were listed in Table S2, which matched the results of the R_Ω. Correspondingly, the description about the electrical conductivity of all the prepared samples was also modified in the revised manuscript. (*Please see page 3, line 28 in the revised version: Electrical conductivity measurements were carried out on cylindrical pellets compressed from the powder samples at 30 MPa between two copper electrodes....followed by conversion to conductivity.; Page 6, line 24 in the revised version: High electrical conductivity of support is favorable to the supported catalysts for the enhancement of catalytic performance. Prior the catalytic performance test, the electrical conductivities of all the prepared samples were measured and the results shown in Table S2....impede the electron transport and decrease the carrier mobility, and reducing electrical conductivity.[43]*)

Q4: Fig.3 and Fig. 6 captions need be checked carefully. It is (a) and (b) rather than (f) and (g) in Fig.3 caption. In Fig. 6, (c) and (d) picture cannot be founded.

Reply: Thanks a lot for the reviewer’s suggestion. According to the suggestions, we have corrected the incorrect description. (*Please see Fig.3 and Fig. 6 captions in the revised manuscript: Fig. 3*

HRTEM images of (a) SnO₂, (b) 40IrO₂/Co_{0.2}Sn_{0.8}O₂. Fig. 6 (a) Cyclic voltammetry curves of 40IrO₂/Co_xSn_{1-x}O₂, pristine SnO₂ and unsupported IrO₂ in N₂-saturated 0.5 M H₂SO₄ solution at a scan rate is 50 mV·s⁻¹; (b) LSV curves of 40IrO₂/Co_xSn_{1-x}O₂, pristine SnO₂ and unsupported IrO₂)

Q5: There are many messy layout and grammatical errors. For example, page 2, line 41, space is missing between number and unit; page 3, line 12, “ml” should be “mL”.

Reply: Thanks a lot for the reviewer’s suggestion. According to the suggestions, we carefully checked the whole manuscript, and made some necessary corrections to the language. (*Please see page 2, the last line in the revised manuscript: “~~20 ml~~” “20 mL”; page 3, line 13: “~~0.12g~~” “0.12 g”; page 4, line 1, “~~2 ml~~” “2 mL”; page 5, line 14: It is noted that the particles sizes of Co_xSn_{1-x}O₂ supports have a gradually decrease with Co doped content increased; page 7, line 22: The potentials at the current density of 10 mA cm⁻² are listed in Table 2. The measured potentials are 1.577, 1.570, 1.557, 1.541 and 1.554 V vs. RHE for unsupported IrO₂; page 9, line 9: the enhancement of electrical conductivity and the higher charge transfer; Page8, line 18: It can be seen that the unsupported IrO₂ shows a lower R_Ω (~~76 mΩ cm²~~, 63.1) because of the excellent electrical conductivity than 40IrO₂/Co_xSn_{1-x}O₂. The R_{ct} value of 40IrO₂/Co_{0.2}Sn_{0.8}O₂ is 32 mΩ cm² and exhibits the lowest ohmic-resistance charge transfer resistance; The caption of Table 1 in the revised figures: ~~The calculated grain sizes of the prepared samples.~~ The lattice constant, grain sizes, BET surface area and BJH adsorption average pore diameter results of Co_xSn_{1-x}O₂ (x = 0, 0.1, 0.2, 0.3).; The Table 2: The obtained values of Voltammetric charge (C/g), the potentials at 10 mA cm⁻², cell potential at 1 A·cm⁻², R_Ω and R_{ct} of the prepared samples.*)

If there appears any question, please do not hesitate to contact me. We look forward to hearing from you for your decision at your earliest convenience.

With best wishes,

Gang Chen, Ph. D, Professor
College of Materials Science and Engineering
Hunan University
Changsha 410012, Hunan, China
E-mail: chengang@hnu.edu.cn